# Conformational State of Fenamates at the Membrane Interface: A MAS NOESY Study

**DOI:** 10.3390/membranes13060607

**Published:** 2023-06-17

**Authors:** Ilya A. Khodov, Konstantin V. Belov, Daniel Huster, Holger A. Scheidt

**Affiliations:** 1G.A. Krestov Institute of Solution Chemistry, Russian Academy of Sciences, 153045 Ivanovo, Russia; kvb@isc-ras.ru; 2Institute for Medical Physics and Biophysics, Leipzig University, 04107 Leipzig, Germany; daniel.huster@medizin.uni-leipzig.de

**Keywords:** fenamates, structure, POPC membrane, bilayer interface, NOESY MAS spectra, conformational changes

## Abstract

The present work analyzes the ^1^H NOESY MAS NMR spectra of three fenamates (mefenamic, tolfenamic, and flufenamic acids) localized in the lipid–water interface of phosphatidyloleoylphosphatidylcholine (POPC) membranes. The observed cross-peaks in the two-dimensional NMR spectra characterized intramolecular proximities between the hydrogen atoms of the fenamates as well as intermolecular interactions between the fenamates and POPC molecules. The peak amplitude normalization for an improved cross-relaxation (PANIC) approach, the isolated spin-pair approximation (ISPA) model, and the two-position exchange model were used to calculate the interproton distances indicative of specific conformations of the fenamates. The results showed that the proportions of the A+C and B+D conformer groups of mefenamic and tolfenamic acids in the presence of POPC were comparable within the experimental error and amounted to 47.8%/52.2% and 47.7%/52.3%, respectively. In contrast, these proportions for the flufenamic acid conformers differed and amounted to 56.6%/43.4%. This allowed us to conclude that when they bind to the POPC model lipid membrane, fenamate molecules change their conformational equilibria.

## 1. Introduction

Bilayer assemblies of phosphatidyloleoylphosphatidylcholine (POPC) represent a standard model to mimic cellular membranes and are widely used to study the interactions of small molecules with lipid bilayers [1,2,3,4]. POPC is a common lipid that is present on the external leaflet of human red blood cell membranes [5] and is responsible for constructing fluid membranes at room temperature. Information on the configuration and mobility of drugs in cellular membranes is valuable for the development of therapies against various diseases. These studies mainly focus on determining the features of the conformational lability of membrane-embedded molecules. Due to the highly dynamic structure of lipid membranes, drug molecules [6] can assume many different conformations [7] with low transition barriers between them [8]. This provides the perfect environment for an effective adaptation of small molecular-weight pharmaceutical compounds [7]. This remarkable property of the lipid membrane allows small molecules to be effectively separated, enriched, and redistributed in the lipid bilayer; furthermore, due to the domain structure, they can target membrane proteins such as receptors, proteases, or channels. Therefore, it is important to study the interaction of membranes with small drug molecules to better understand their mechanism of action.

The effect of the conformation of small molecules on bilayer structures is rarely considered in the literature. In [9], a group of scientists, led by Iván Ortega-Blake and using the example of cholesterol and ergosterol, showed the effect of the conformation of the injected molecule on the structure of the lipid membrane, which was modeled based on POPC and egg sphingomyelin. It has been established that the conformation of small molecules localized in the bilayer is one of the main factors influencing the structure and dynamics of the domains present in the bilayer. Another study presented by Markus Zweckstetter et al. [10] showed that the conformation of α-synuclein determined its binding to synaptic vesicles and, in some cases, could lead to the development of Parkinson’s disease. In their work, the authors tested various small molecules for their effectiveness in stabilizing the required conformation of α-synuclein. The authors modeled the vesicles’ surface using POPC and 1-palmitoyl-2-oleoyl phosphatidic acid (POPA). Our work is a logical continuation of previous studies [11] related to determining the position and orientation of small fenamate molecules in the POPC lipid membrane and finding its order parameter. Fenamates are a class of non-steroidal anti-inflammatory drugs (NSAIDs) used to treat pain and inflammation. The molecules of fenamates are derivatives of anthranilic acid and can potentially reprofile drug forms. These NSAIDs treat various conditions, including arthritis, menstrual cramps, and headaches. Common fenamate drugs include mefenamic acid, tolfenamic acid, and flufenamic acid [12,13,14]. Moreover, a series of studies [15,16,17,18,19] aimed to establish the distribution of the proportions of fenamate conformer groups in a solution of dimethyl sulfoxide (DMSO-*d*_6_) as well as in a mixed solvent based on supercritical carbon dioxide (scCO_2_). It was found that flufenamic acid significantly differed in conformational preferences compared with mefenamic and tolfenamic acids in DMSO-*d*_6_ and scCO_2_. Encouraged by the works of Ortega-Blake, Zweckstetter, and their co-authors, an attempt was made to determine the predominant fenamate conformation in the POPC membrane.

We used an approach based on nuclear Overhauser effect spectroscopy (NOESY) under magic-angle spinning (MAS) [20,21,22] conditions for this purpose, considering the efficiency previously shown of the NOESY method in studying the conformational preferences of small molecules in solutions [23,24,25,26,27,28,29,30,31], which made it the most appropriate.

The log *p*-values of the fenamates were 4.77 for MFA, 5.00 for TFA, and 4.84 for FFA (obtained from https://www.molinspiration.com accessed on 2 June 2023), demonstrating that the drugs’ unionized forms were highly lipophilic [32]. The logP values suggested that fenamates could be found in membranes and could interact with the lipids, as also shown by the intense cross-peak NOESY between protons in fenamates and lipids [11]. Consequently, we predicted that the fenamates could pass into the bilayers of cells. Multi-lamellar lipid vesicles (MLVs) based on POPC offer an effective model for cell membranes as they have low curvature and permit the characterization of the membrane interaction of small molecular-weight drugs [33,34]. It is worthwhile to study the conformation of fenamates in a membrane environment as these molecules are highly lipophilic. It is well known that small lipophilic molecules first partition into the membrane, where the diffusion is two-dimensional rather than three-dimensional (as in a solution), which increases the effective concentration of these molecules [35] and increases the likelihood of binding with the target protein. Our study was performed with a rather simplistic POPC membrane, which is often required for the experimental method of NOESY MAS [36]. When using more realistic membrane models, NMR peaks are often poorly resolved and it is then impossible to determine the structure of the molecules. In particular, cholesterol significantly broadens the NMR lines [37,38], rendering NOESY experiments impossible.

## 2. Materials and Methods

Commercial compounds were used for all NMR experiments. POPC was synthesized by Avanti Polar Lipids, Inc. (Alabaster, AL, USA); mefenamic (2-[bis(2,3-dimethylphenyl)amino]benzoic acid) (CAS No. 61-68-7), tolfenamic (2-[bis(3-chloro-2-methylphenyl)amino]benzoic acid) (CAS No. 13710-19-5), and flufenamic (2-[[3-(trifluoromethyl)phenyl]amino]benzoic acid) (CAS No. 530-78-9) acids were purchased from Sigma-Aldrich (St. Louis, MO, USA). All compounds were used without further purification. 

For the sample preparation for the NMR measurements, a 4:1 mixture of POPC with fenamate was dissolved in trichloromethane, followed by evaporation of the solvent. The 4:1 ratio of POPC to fenamate was chosen in these experiments to obtain sufficient signal intensities in the NOESY spectra. The resulting lipid films, which were lyophilized overnight under a high vacuum after being redissolved in cyclohexane and frozen in liquid nitrogen [39,40], were hydrated with 50 wt% D_2_O for magic-angle spinning (MAS) ^1^H NMR experiments.

The morphology of the MLVs of the membranes was confirmed by ^31^P NMR spectroscopy, as shown in [11], which can easily differentiate between lipid bilayers as well as hexagonal and isotropic structures. 

To conduct 2D ^1^H MAS NOESY experiments [41,42], the obtained MLV samples were homogenized by ten freeze–thaw cycles in a 40 °C water bath and in liquid nitrogen. A Bruker Avance III 600 MHz NMR spectrometer (Bruker Biospin GmbH; Rheinstetten, Germany) was used to obtain the ^1^H MAS NMR spectra. The resulting homogeneous samples were placed in a 4 mm MAS rotor and measured. The NMR spectra were recorded by rotating the sample at the magic angle with a frequency of 6 kHz and 90° pulse lengths of 4 μs. The mixing times for recording the 2D ^1^H MAS NOESY spectra were 0.1, 100, 200, 300, and 500 ms. The number of data points in recording the spectra were 216 and 4096 along the F1 and F2 axes, respectively, the number of scans was 40, and the combined value of the relaxation delay and mixing time was 3.2 s. The ^1^H MAS NMR experiments were conducted at 30 °C. The integration of the diagonal peaks and cross-peaks was achieved using Bruker Topspin 3.6.3 software, from which the spin-pair model fitting [8,22] of the cross-peak volume over the mixing time was applied using Origin (OriginLab Cooperation, Northampton, MA, USA) to obtain the cross-relaxation rate values.

## 3. Results

The chemical structure of fenamate molecules [43,44,45,46,47] was represented by an anthranilic acid derivative in which one of the aromatic fragments attached to the nitrogen atom was replaced by 2,3-dimethylphenyl (MFA), 2-methyl-3-chlorophenyl (TFA), and 3-(trifluoro-methyl) phenyl (FFA) fragments (see Figure 1).

Different conformers of fenamates molecules were associated with a change in the value of the dihedral angle *τ_1_* (C2–N(H)–C3–C7). Thus, for conformers A and C of fenamates, the position of the aromatic rings corresponded with that shown in Figure 1. The value of *τ_1_* (C2–N(H)–C3–C7) ≈ −135° (A and C), and the substituents X and Y were co-directed with the carboxyl group of the anthranyl fragment. At the same time, for conformers B and D, the position of the aromatic fragment with substituents X and Y was opposite to the carboxyl group and the value of *τ_1_* ≈ −76° (see Appendix A, Figure A1).

The current work analyzed the ^1^H NOESY MAS NMR spectra of three fenamates at the bilayer interface (see Appendix A, Figure A2). The observed cross-peaks in the two-dimensional spectrum characterized the intramolecular and intermolecular interactions of the respective hydrogen atoms of fenamates and POPC. The chemical shifts of the signals of the NMR belonging to the protons of the CH_3_ groups of the fenamate molecules (0–3 ppm) were well consistent with the data from organic environments [16] located in the field and overlapped with the intensive atoms of hydrogen POPC molecules. The observed cross-peaks in the spectral region from 6 ppm to 9 ppm corresponded with the CH groups of aromatic moieties of the fenamate molecules. The appendices provide a full list of the ^1^H NMR signals that were assigned to all fenamates.

The magnetization exchange observed in lipid membranes is largely due to dipolar cross-relaxation between two nearby proton spins, and this process can be quantified using the cross-relaxation rate *σ_IS_*. This rate is strongly affected by the distance between the two spins, *r_IS_*, and the correlation time of the molecular motion *τ_c_*.
(1)$$\sigma_{IS}=\frac{\hbar^{2}\mu_{0}^{2}\gamma^{4}\tau_{c}}{{40\pi }^{2}r_{IS}^{6}}\left\{-1+\frac{6}{1+4\omega_{0}^{2}\tau_{c}^{2}}\right\}$$

In Equation (1), ħ is the ratio of Planck’s constant divided by 2*π*, the magnetic constant (*µ*_0_), the Larmor frequency (*ω*_0_), and the gyromagnetic ratio (γ). As a result of the fact that the cross-relaxation rate is very dependent on distance, only interactions within a distance of 5 Å contribute to the cross-relaxation rate.

The peak amplitude normalization for an improved cross-relaxation (PANIC) approach [23,48,49,50], the isolated spin-pair approximation (ISPA) model [51,52,53,54], and the two-position exchange model [23] were used to calculate the proportions of fenamate conformer groups in a lipid membrane modeled by POPC. The volume of the cross-peak between two spin systems I and S in the spin-pair interaction model could be determined by the formula in [55]. This model assumes that the spin systems are adequately decoupled so that the entire multi-spin system can be reduced to a single pair of spins.
(2)$$A_{IS}\left(t_{m}\right)=\frac{A_{II}\left(0\right)}{2}\left[1-exp(-2\sigma_{IS}t_{m})\right]\times exp\left(-\frac{t_{m}}{T_{1,IS}}\right)$$
where *A_IS_*(*t_m_*) is the cross-peak volume at the time of mixing tm, *A_II_*(0) is the diagonal peak volume at the first moment of mixing *t_m_* = 0, and *T*_1,*IS*_ defines the amount of magnetization that has leaked into the lattice.

The spin-pair model requires only the integration of the cross-peaks between the intramolecular protons to calculate their respective cross-relaxation rates. This approach can save time and effort when only a few cross-relaxation rates are needed, as was the case here. Not having all of the information from the complete relaxation matrix [8] is acceptable when evaluating the preferred conformers within the membrane matrix as monitoring all magnetization transfer processes is unnecessary.

The PANIC approach has been proven to increase the linearity of the initial buildup rate in NOESY and exchange experiments when the ratio of the cross-peak to the diagonal peak is taken into account. This enables more data points that can be acquired with longer mixing times to be included in the analysis, ultimately leading to improved accuracy in the cross-relaxation rates and internuclear distances.

One of the most suitable characteristics indicating a change in the conformation of a molecule is the value of the internuclear distances. In the case of fenamates, when the conformation changed from A+C to B+D (see Figure 2), the distance values between the protons of cyclic fragments of the molecules changed. Within the framework of the proposed approaches, it was necessary to have information about the values of the reference and experimental distances to determine the proportions of the fenamate conformer groups in the systems under study. The value of the reference distance was retained for all conformers under consideration within the experimental error (±0.11 Å). At the same time, the value of the observed distance changed with a change in the conformation of the molecule. The choice of the reference distance (H6–H11/12) was based on the literature [11,15,16,17], and its value was 2.76 ± 0.11 Å for all conformers of the studied molecules. The experimental cross-relaxation rates (*σ*) were determined using the ISPA model. The calculation of the distance values from the NOESY cross-relaxation rates using Equation (3) below showed three different types of interactions between hydrogen atoms. Among them were interactions between neighboring hydrogens of CH groups within one benzene ring, the distances for which were ≈2.47 Å (see Figure 2; red line). The distances between the non-adjacent hydrogens of the CH groups in one benzene ring were 4.28 Å and greater (see Figure 2; blue line). Interactions between the hydrogens of the different benzene rings (see Figure 2; green line) were also suitable to identify the individual conformers (see Table A1). The most probable dimers—which should have been present in the crystal, according to the Cambridge Crystallographic Database [56,57,58,59,60,61,62,63,64,65,66,67]—provided distances between protons of the cyclic fragments from 5.3 Å to 6.9 Å, leading to the fact that the contribution to the NOE intensity from such interactions was not experimentally observed.

We obtained the distances between H9/10 and H11/12 for the fenamate molecules (Figure 2) in POPC as follows:(3)$$r_{ij}^{exp}=r_{0}\sqrt[{6}]{\frac{\sigma_{0}^{exp}}{\sigma_{ij}^{exp}}}$$
where $r_{ij}^{exp}$ is the value of the distance obtained from the experimental data, *r*_0_ is the reference distance H6–H11/12 (2.76 ± 0.11 Å) obtained from data from the Cambridge Crystallographic Database, *σ*_0_ is the value of the cross-relaxation rate for the reference distance obtained from the NOESY data, and *σ_ij_* is the cross-relaxation rate for the experimental distance also obtained from the experimental NOESY data.

The experimental interproton distances were computed using both cross-relaxation rates *σ*_0_ and *σ_ij_* for the reference and conformation-dependent distances, respectively, as demonstrated in Equation (3). These values were obtained from the analysis of the dependence of the normalized values of the integrated cross-peak intensities on the mixing time in the framework of the initial rate approximation (IRA), as shown in [15,23,68].

Thus, according to the data of quantum chemical calculations, the distances H9/10–H11/12 for the A+C and B+D conformer groups ($r_{A+C}^{calc.}$ and $r_{B+D}^{calc.}$) were 3.12 Å and 4.62 Å, 3.00 Å and 4.76 Å, and 2.92 Å and 5.33 Å for mefenamic, tolfenamic, and flufenamic acids, respectively. At the same time, the distances H9/10-H11/12 according to the experimental data ($r_{ij}^{exp}$) were 3.47 Å, 3.35 Å, and 3.20 Å for the same objects of study. The cross-relaxation rates ($\sigma_{ij}^{exp}$) were 2.17 × 10^−3^ s^−1^, 1.89 × 10^−3^ s^−1^, and 0.71 × 10^−3^ s^−1^, respectively.

Based on the experimental data and quantum chemical calculations, using Equation (4) within the framework of the two-position exchange model, the proportions of the A+C and B+D conformer groups of fenamate molecules in the POPC membrane were calculated (see Figure 3).
(4)$$\frac{1}{{\left(r_{ij}^{exp}\right)}^{6}}=\frac{x_{A+C}}{{\left(r_{A+C}^{calc.}\right)}^{6}}+\frac{x_{B+D}}{{\left(r_{B+D}^{calc.}\right)}^{6}}$$
where *x_A+C_* and *x_B+D_* are the percentages of the A+C and B+D group of conformers, respectively, $r_{A+C}^{calc.}$ is the value of the conformationally determined distance (H9/10-H11/12) of the conformers of the A+C group, and $r_{B+D}^{calc.}$ is the value of the conformationally determined distance (H9/10-H11/12) of the conformers of the B+D group. Figure 3 shows the results.

## 4. Discussion

Figure 3 shows that the proportions of the A+C and B+D conformer groups of mefenamic and tolfenamic acids in POPC membranes were comparable within the experimental error, taking into account that the error in determining the distance in the experiment reached 2–3% (see Appendix A, Figure A3); in the NOESY experiments, they were indistinguishable and amounted to 47.8%/52.2% and 47.7%/52.3%. At the same time, the proportions of the flufenamic acid conformer groups differed and amounted to 56.6%/43.3%. Thus, for MFA and TFA, the predominant conformation was B+D; for FFA, it was A+C. In addition, comparing the obtained data with the results of other studies [15,16,17], it should be emphasized that the predominant conformation for all considered fenamates drastically differed from what was observed in DMSO-*d*_6_, except for FFA. In the previously considered systems for MFA and TFA, the group of conformers A+C was predominant; for FFA, it was the group of conformers B+D. This emphasized the important impact of the POPC model membrane, where fenamate molecules changed their three-dimensional structure. It is important to note that in the series MFA, TFA, and FFA in the presence of POPC, it was for FFA that the distribution of proportions of conformer groups differed, as did the value of the lipid chain order parameter measured and discussed in [11]. Probably due to the peculiarities of the chemical structure [17], FFA had its specifics of interaction with POPC; as a result, the changes in the three-dimensional structure observed were not characteristic for other the considered fenamates. As shown in a recent study [69], fluorinated fragments in the structure of the molecules of active pharmaceutical ingredients were decisive when changing the forms’ pharmaceutical and physical properties. The conformation distribution did not change much between DMSO-*d*_6_ and POPC; this could have been due to other factors such as the size and shape of the FFA that allowed it to interact more strongly with the POPC molecules. On the other hand, MFA and TFA may not have had the same properties, which was why the conformation distribution changed more when moving from DMSO-*d*_6_ to POPC. We noted that the modulation of the logP (octanol–water partition coefficient) by fluorination was associated with a change in the permeability of the concomitant membrane modeled in this work and based on POPC. Thus, the results obtained in this study may help in the search for further modifications to repurpose the dosage forms of several fenamates.

## 5. Conclusions

We studied the structure of fenamate molecules—mefenamic, tolfenamic, and flufenamic acids—in POPC membranes using MAS NOESY data. We found that when embedded in POPC membranes, the fenamate molecules changed their three-dimensional structure. For MFA and TFA, the predominant conformation was B+D; for FFA, it was A+C. Moreover, as shown in a previous study [11], the localization of FFA in the POPC lipid membrane differed from the localization of MFA and TFA. Probably due to its chemical structure, FFA had its specifics of interaction with POPC; as a result, it changed its structure to a conformation that was not in the other fenamates. This study showed that the conformation of the fenamates in a membrane environment could be used to predict their behavior when interacting with cellular membranes. Furthermore, differences in the conformation distribution between DMSO-*d*_6_ and POPC further suggested that FFA might interact more strongly with POPC molecules, whereas MFA and TFA might not have the same properties. This research provides an insight into the conformation of fenamates in a membrane environment and may help future research on drug delivery and interactions with cell membranes.

## Figures and Tables

**Figure 1 membranes-13-00607-f001:**
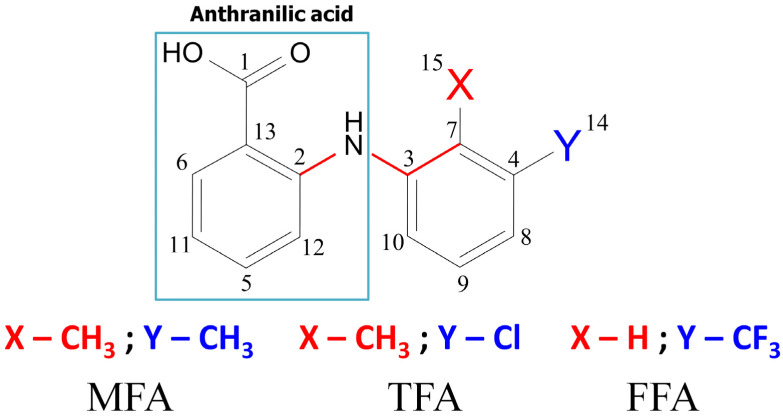
Structure of fenamate molecules, where X and Y are positions of CH_3_ and CH_3_, and Cl, H, and CF_3_ are substituents for mefenamic (MFA), tolfenamic (TFA), and flufenamic (FFA) acids, correspondently.

**Figure 2 membranes-13-00607-f002:**
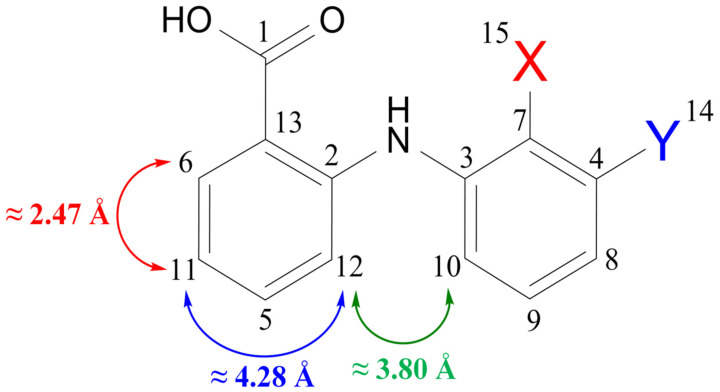
Structure of fenamate molecules with indication of different types of interproton interactions observed in the NOESY experiment. Interactions between adjacent hydrogens of CH groups within the same benzene ring (red line), noncontiguous hydrogens of CH groups (blue line), and between hydrogens in different benzene rings (green line).

**Figure 3 membranes-13-00607-f003:**
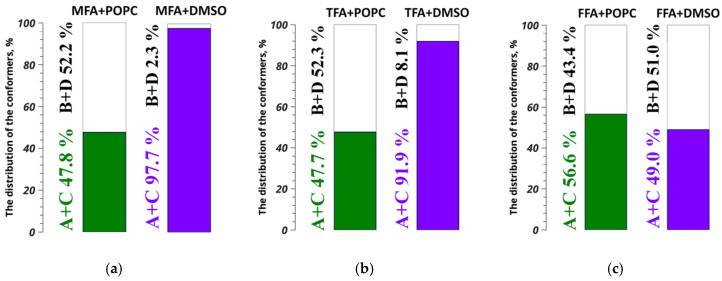
Distribution diagrams of the relative proportions of the A+C (green) and B+D (purple) conformer groups for mefenamic (**a**), tolfenamic (**b**), and flufenamic (**c**) acids in POPC membranes.

## Data Availability

Not applicable.

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
