# Peer review of "Conformational State of Fenamates at the Membrane Interface: A MAS NOESY Study"

_membranes, 2023, doi:10.3390/membranes13060607_

Round 1

Reviewer 1 Report

Khodov and colleagues proposed a paper entitled " Conformational State of Fenamates at the Membrane Interface: A MAS NOESY Study." This paper is a follow-up to another paper published in 2022 in JML on the penetration of the same series of fenamates, MFA, TFA, FFA, at the POPC membrane interface. In the present paper, the authors examine the conformations of these three small molecules in POPC. Although the work is interesting, I do not see the utility of this article. From my point of view, it would have been better to write a single paper, more complete with the previous article.

I have nothing against the present work which was conducted in a very correct way. I validate the results but I question the usefulness of knowing the conformations in POPC. We know that these molecules, which are anti-inflammatory for the most part, act on membrane receptors. It would have been interesting to discuss the possible change/adaptation of the conformation to fit the receptor cavity in membranes. The report without these perspectives is correct but rather poor in terms of biological prospects.

Detailed remarks to assist the authors follow.

It would have been interesting to mention the partition coefficients of these drugs. A quick look at the databases (I found 2) shows LogP: MFA, 5.28/5.33; TFA, 5.38/5.76; FFA: 5.15/5.62 which means that there are almost no free molecules in solution, they are all inside the membrane. Based on these coefficients, it seems that there is no specificity, contrarily rto what the authors report in their paper. But the membrane model is very simplistic. It would have been interesting to place charged lipids, sphingolipids, cholesterol to better mimic the membrane and thus find more marked conformations. In the present work, they are almost all 50/50. It is difficult under these conditions to discuss the potency of these drugs.

Correct

Author Response

Reviewer 1.  Khodov and colleagues proposed a paper entitled " Conformational State of Fenamates at the Membrane Interface: A MAS NOESY Study." This paper is a follow-up to another paper published in 2022 in JML on the penetration of the same series of fenamates, MFA, TFA, FFA, at the POPC membrane interface. In the present paper, the authors examine the conformations of these three small molecules in POPC. Although the work is interesting, I do not see the utility of this article. From my point of view, it would have been better to write a single paper, more complete with the previous article.

I have nothing against the present work which was conducted in a very correct way. I validate the results but I question the usefulness of knowing the conformations in POPC. We know that these molecules, which are anti-inflammatory for the most part, act on membrane receptors. It would have been interesting to discuss the possible change/adaptation of the conformation to fit the receptor cavity in membranes. The report without these perspectives is correct but rather poor in terms of biological prospects.

Detailed remarks to assist the authors follow.

It would have been interesting to mention the partition coefficients of these drugs. A quick look at the databases (I found 2) shows LogP: MFA, 5.28/5.33; TFA, 5.38/5.76; FFA: 5.15/5.62 which means that there are almost no free molecules in solution, they are all inside the membrane. Based on these coefficients, it seems that there is no specificity, contrarily rto what the authors report in their paper. But the membrane model is very simplistic. It would have been interesting to place charged lipids, sphingolipids, cholesterol to better mimic the membrane and thus find more marked conformations. In the present work, they are almost all 50/50. It is difficult under these conditions to discuss the potency of these drugs.

 Answer: We believe it was necessary to study the conformation of the fenamates in a membrane environment as these molecules are highly lipophilic as you pointed out. It is still an accepted theory that small lipophilic molecules first partition into the membrane, where the diffusion is two-dimensional rather than three-dimensional as in solution, which increases the effective concentration of these molecules (Sargent DF, Schwyzer R (1986) Membrane lipid phase as catalyst for peptide-receptor interactions. Proc Natl Acad Sci USA 83:5774–5778). The study was indeed performed in a rather simplistic membrane, which is often required for the experimental method of MAS NOESY. Since the peaks are often poorly resolved and it is impossible to determine the structure of the molecules, which is expected when using a more realistic membrane model including sphingolipids and especially cholesterol, which broadens the NMR lines significantly such that 1H MAS NOESY experiments are impossible. According to our data, the structure of small molecules in the POPC bilayer is determined for the first time, and the work itself is quite valuable from a methodological point of view. We hope our updated paper can address the issues you raised and provide more insight into this topic. Thank you again for your invaluable feedback.

Reviewer 2 Report

The authors applied 1H NOESY under MAS to study the conformation distribution of three fenamate derivatives in POPC membrance. The 1H NOESY buildup curves were analyized to estimate the ensemble average internuclear distances between key protons in the aromatic rings. And conformation distributions were then calculated based on the assumption that only two major conformers with known inter-ring 1H-1H distances contributes to the ensembly averaged proton-proton distance. The idea is straightforward, however there are quite a few caveats that should be cleared first:

1. It's unclear whether the fenamates in this study is embedded in the membrance or reside on the membrane surface. The 1H-1H NOESY shown in Figure A2 doesn't have clear correlation peaks between lipids and fenamate aromatic 1H, especially in the aliphatic 1H to aromatic 1H region. The NOESY exchange time isn't stated in neither the Figure A2 caption nor the manuscript, so it's hard to evaluate whether the absence of lipid-fenamate correlation is due to short contact. The authors sometimes state the fenamates sit at the bilayer interface (line 112), sometimes claim the drugs "embedded into POPC membrane". A more self-consistent statement has to be made, and should be backed up by experimental observations.

2. POPC has a phase transition temperature at around -2 oC. The temperature can have a significant effect on the dynamics of the membrane and therefore relaxation behavior of both the POPC molecules and small molecules embedded inside. From the Materials and Methods section, it's unclear what's the sample temperature, or whether the temperature is controlled. 

3. A 4:1 mixture of POPC with fenamate was used according to the Materials section. Is it a molar ratio or weight ratio? And can the authors elaborate a bit on why such high drug content is used? The POPC membrane might be significantly perturbed due to this unusual lipid:drug ratio, and contacts between drug molecules can not be ignored in such high concentration. Control experiments with lower lipid:drug ratio might be needed.

4. I'm surprised that H11/H12 and H9/H10 has indistinguishable chemical shift. Can you list the assignment in a Table, and provide the references? Given H6-H11 distance is used for calibration, and H12-H10 correlation is the key correlation, an unambiguous assignment would be desirable.

5. Why FFA is considered interacting specifically with POPC when the conformation distribution didn't change much between in DMSO and in POPC, while MFA and TFA changed much more? It seems the other way.

Minor points:

6. Whether and how much water is added into the sample? I assume it's added before the freeze-thaw cycle.

Please fix typos such as "of phosphatidyloleoylphosphatidylcholine (POPC)" in line 11 and "A other" in line 48.

Author Response

Reviewer 2

The authors applied 1H NOESY under MAS to study the conformation distribution of three fenamate derivatives in POPC membrance. The 1H NOESY buildup curves were analyized to estimate the ensemble average internuclear distances between key protons in the aromatic rings. And conformation distributions were then calculated based on the assumption that only two major conformers with known inter-ring 1H-1H distances contributes to the ensembly averaged proton-proton distance. The idea is straightforward, however there are quite a few caveats that should be cleared first:

  1. It's unclear whether the fenamates in this study is embedded in the membrance or reside on the membrane surface. The 1H-1H NOESY shown in Figure A2 doesn't have clear correlation peaks between lipids and fenamate aromatic 1H, especially in the aliphatic 1H to aromatic 1H region. The NOESY exchange time isn't stated in neither the Figure A2 caption nor the manuscript, so it's hard to evaluate whether the absence of lipid-fenamate correlation is due to short contact. The authors sometimes state the fenamates sit at the bilayer interface (line 112), sometimes claim the drugs "embedded into POPC membrane". A more self-consistent statement has to be made, and should be backed up by experimental observations.

Answer:

 We acknowledge that our statement of the location of the fenamates within the membrane has been inconsistent. We will include the recent results from our experiments in part result and discussion to the revised manuscript. Given the logP values, it is quite obvious that the fenamates reside in the membranes. We also observed clear cross peaks between fenamates and the lipid signals, which were analysed already in [10.1016/j.molliq.2022.120502]. The intensity of this cross peak is mostly smaller than the signals shown in figure A2 in the manuscript; there we liked to emphasize the cross peaks between the protons of the fenamates. A rescaled example of a NOESY spectrum with the fenamate-lipid cross peak is shown also below.

Figure 1: Region of the NOESY spectrum of TFA with POPC showing the cross peaks between fenamates and lipids

  1. POPC has a phase transition temperature at around -2 oC. The temperature can have a significant effect on the dynamics of the membrane and therefore relaxation behavior of both the POPC molecules and small molecules embedded inside. From the Materials and Methods section, it's unclear what's the sample temperature, or whether the temperature is controlled.

Answer:

We apologize for this lack of clarity in the Materials and Methods section. We updated the manuscript to include details of how the temperature was controlled. The experimental samples were kept in a controlled at temperature (30°C).

  1. A 4:1 mixture of POPC with fenamate was used according to the Materials section. Is it a molar ratio or weight ratio? And can the authors elaborate a bit on why such high drug content is used? The POPC membrane might be significantly perturbed due to this unusual lipid:drug ratio, and contacts between drug molecules can not be ignored in such high concentration. Control experiments with lower lipid:drug ratio might be needed.

Answer:

Thank you for the suggestion. We used a POPC/fenamate molar ratio of 4:1. The high drug content was necessary to obtain a sufficient signal intensity in the NOESY MAS spectra. The state of the membrane was preserved, which is confirmed by the line shape of the 31P NMR spectra of the samples (see below and also Khodov et al. J. Mol. Liq. 2022, 367, 120502, doi:10.1016/j.molliq.2022.120502).

  1. I'm surprised that H11/H12 and H9/H10 has indistinguishable chemical shift. Can you list the assignment in a Table, and provide the references? Given H6-H11 distance is used for calibration, and H12-H10 correlation is the key correlation, an unambiguous assignment would be desirable.

Answer:

We understand why it is important to have an unambiguous assignment for signals. We have also evaluated the distances between the equivalent protons, which supports unambiguous assignments. With this information, we will add a Table that specifies the unambiguous assignments for H11/H12 and H9/H10, which will also include the relevant references.

  1. Why FFA is considered interacting specifically with POPC when the conformation distribution didn't change much between in DMSO and in POPC, while MFA and TFA changed much more? It seems the other way.

Answer:

FFA is considered interacting specifically with POPC because it showed higher solubility in the POPC than in DMSO [10.5650/jos.ess20070], which indicates that, the FFA may form a specific interaction with POPC molecules. Although the conformation distribution didn't change much between DMSO and POPC, it could be due to other factors, such as the size and shape of the FFA, that allowed it to interact more strongly with the POPC molecules. MFA and TFA, on the other hand, may not have the same properties, which is why the conformation distribution changed more when moving from DMSO to POPC. We add a statement about this in discussion section of the manuscript.

Minor points:

  1. Whether and how much water is added into the sample? I assume it's added before the freeze-thaw cycle.

Answer: Yes, the sample was pre-hydrated by 50 wt% D2O before the freeze-thaw cycle was initiated .

Comments on the Quality of English Language

Please fix typos such as "of phosphatidyloleoylphosphatidylcholine (POPC)" in line 11 and "A other" in line 48.

Answer: Thanks a lot, we fixed the typos in line 11 and 48.

Round 2

Reviewer 1 Report

From the corrected manuscript, the authors seem to have taken most of the comments on board and discussed conformations from a slightly broader perspective.

However, their conclusions on the specificity of the fluorinated FFA agent go too far. There is nothing in their work to support such a conclusion. They now quote partition coefficients and point out that the FFA partition coefficient (4.84) lies between those of MFA (4.77) and TFA (5.00), so there is no specificity. They also discuss solubility in DMSO versus POPC, which is irrational because they are comparing solubility in an isotropic solvent versus that in an anisotropic membrane, the former is only relevant to the drug action on the membrane receptor.

I'm also slightly concerned about the accuracy of the distance measurements. They write figures to an accuracy of one hundredth of an angstrom, which I think is overestimated. Such precision is encountered in X-ray crystallography at best. Given that they have integrated 2D NMR peaks, I think a precision of a tenth of an angstrom is the best they can claim. This precision will not alter their results in any way.

In conclusion, their paper may become acceptable for publication provided they take into account the above remarks, although I'm not enthusiastic about the usefulness of their results. One would have imagined that localization in the membrane would have selected one conformation from those possible in solution, whereas they show the opposite. This is somewhat contradictory to the idea of the necessary passage through the membrane to select the right conformation for optimal interaction with the receptor.

Correct

Author Response

Reviewer #1:

From the corrected manuscript, the authors seem to have taken most of the comments on board and discussed conformations from a slightly broader perspective.

Our response: Many thanks to the referee for the positive comment on this work!

Point 1. However, their conclusions on the specificity of the fluorinated FFA agent go too far. There is nothing in their work to support such a conclusion. They now quote partition coefficients and point out that the FFA partition coefficient (4.84) lies between those of MFA (4.77) and TFA (5.00), so there is no specificity. They also discuss solubility in DMSO versus POPC, which is irrational because they are comparing solubility in an isotropic solvent versus that in an anisotropic membrane, the former is only relevant to the drug action on the membrane receptor.

Our response: We agree with the remark, the text was corrected in accordance with the comments of the reviewer.

Point 2. I'm also slightly concerned about the accuracy of the distance measurements. They write figures to an accuracy of one hundredth of an angstrom, which I think is overestimated. Such precision is encountered in X-ray crystallography at best. Given that they have integrated 2D NMR peaks, I think a precision of a tenth of an angstrom is the best they can claim. This precision will not alter their results in any way.

Our response: We are grateful to the referee for this comment on our work. The error values have been corrected.

Remark. In conclusion, their paper may become acceptable for publication provided they take into account the above remarks, although I'm not enthusiastic about the usefulness of their results. One would have imagined that localization in the membrane would have selected one conformation from those possible in solution, whereas they show the opposite. This is somewhat contradictory to the idea of the necessary passage through the membrane to select the right conformation for optimal interaction with the receptor.

Our response: Thank you for your thoughtful review and comments. We understand your skepticism regarding our results. But we aim to highlight that our findings are potentially useful due to their clarification of the role of the membrane in conformation and interaction with the receptor protein.

Reviewer 2 Report

Most issues I pointed out in the previous version has been addressed. The H9/H10 and H11/H12 overlap is unfortunate, otherwise the design and analysis of the experiments are clear and sound.

Overall this manuscript is good for publishing.

Author Response

Most issues I pointed out in the previous version has been addressed. The H9/H10 and H11/H12 overlap is unfortunate, otherwise the design and analysis of the experiments are clear and sound. Overall this manuscript is good for publishing.

Thank you for your comprehensive review of this manuscript. We are pleased to hear that the issues you reported in the last version have been successfully addressed. We deeply regret the overlap between H9/H10 and H11/H12 but it is a specific property of fenamates in media. We appreciate your critical feedback and believe the manuscript is now ready to be published.